# CLIP-LAD: Unleash the Potential of CLIP for Few-shot Logical Anomaly Detection

## Abstract

Anomaly detection (AD) is crucial for visual inspections, and includes two main types: structural and logical anomalies. Despite growing interest in AD, most methods focus on structural anomalies, while few works address logical anomaly detection (LAD), which requires a global understanding of the context. Leading LAD methods often advocate segmentation algorithms to parse logical relations within images, necessitating extensive training images or elaborate labels, but they undergo significant performance degradation in low-data scenarios at a risk of over-fitting. This study explores a practical yet challenging scenario where only few-shot normal images are available. To the end, we introduce CLIP-LAD, a novel, training-free method for few-shot LAD. We propose a coarse-to-fine segmentation process, involving foreground extraction and fine-grained alignment, to progressively harness the CLIP's generalization abilities for LAD. Specifically, we first aggregate visual features into different regions with clear boundaries, benefited from the strong visual coherence in vision transformer (ViT), and leverage coarse prompts to help identify the foreground. Within the foreground, we further conduct per-pixel fine-grained classification with fine prompts to parse different parts of an object. The anomaly scoring is derived from the class histograms in the precise segmentation masks. For comprehensive evaluation, we build up a few-shot LAD benchmark based on the MvTec-LOCO dataset and include a series of comparison methods. Experiments on this benchmark demonstrates our superiority in low-data regime.

## 1 Introduction

Anomaly detection is a fundamental yet challenging problem, which entails identifying anomalous patterns that deviate significantly from the training distribution. Specifically, in visual inspection, defects can be broadly classified into structural and logical anomalies (Bergmann et al., 2022). Structural anomalies refer to deviations in the visual structure, texture, or shape of an object from its expected norm, *e.g.*, cracks or scratches. In contrast, logical anomalies emphasize violations of logical constraints or expectations, such as missing, surplus objects, or misplacement.

While AD has recently garnered significant research interest, the majority of these methods (Bergmann et al., 2020; Wang et al., 2021; Li et al., 2024) are biased towards identifying structural anomalies, with limited focus on logical anomaly detection (LAD), which necessitates an understanding the global context beyond patch-level visual analysis. Current LAD approaches can be classified into two categories: feature-based and segmentation-based. Feature-based methods implicitly capture the intricate logical dependencies within images through either knowledge distillation (Batzner et al., 2024) or image reconstruction (Yang et al., 2023). Here, the discrepancy between the original input and its reconstructed counterpart serves as an anomaly indicator. Conversely, segmentation-based methods explicitly infer the relationships among objects or their parts through segmentation (Kim et al., 2024; Liu et al., 2023), delivering enhanced performance due to the granular part-level analysis. Unfortunately, all these LAD methods involve a training process and tend to be data-intensive (Li et al., 2024; Batzner et al., 2024) or require elaborate per-pixel annotations (Kim et al., 2024). This dependency on the data scale and labels poses a significant risk of over-fitting, particularly in low-data regime.

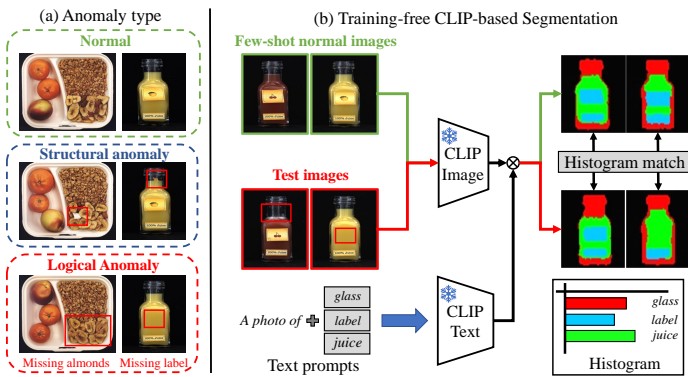

Figure 1: (a) Two types of anomalies: structural anomalies exhibit obvious visual discrepancies compared to normal images, while logical anomalies represent violations of logical constraints. (b) Sketch of our method: we leverage the generalization abilities of CLIP vision-language alignment to parse normality using only a few normal images, and rely on histogram statistic of segmentation for LAD. Anomalies are indicated by red bounding boxes.

In this study, we address the challenging problem of few-shot LAD to meet the requirements of real-world applications, where access to abundant training images is impractical and per-pixel labeling is resource-intensive. Building on the success of segmentation-based methods, we model object/part relationships through segmentation. Motivated by the remarkable results achieved by Contrastive Language-Image Pre-training (CLIP) in open-vocabulary semantic segmentation (OVSS) using image-text pairs, we embrace CLIP to segment regions of interest within the images. However, directly applying CLIP inevitably leads to noisy predictions, especially around the object boundaries, due to the imperfect vision-language alignment at patch level, even with the advanced OVSS method (Hajimiri et al., 2024).

To overcome this issue, we introduce CLIP-LAD, a novel, training-free method to refine the segmentation process for few-shot LAD. We propose a coarse-to-fine segmentation pipeline that consists of foreground extraction and fine-grained alignment, ensuring more accurate object/part segmentation within a few-shot framework. Benefiting from the strong visual coherence in shallow ViT stages, we first cluster the features extracted from these stages into distinct regions. Each of these regions serves as a class-agnostic mask proposal, which is utilized to aggregate patch tokens from the final ViT stage, representing the embedding of the corresponding region. This embedding is then matched with pre-set coarse prompts to distinguish the foreground from the background. Within the identified foreground region, we further perform fine-grained visual-language matching using fined prompts at patch level, ultimately yielding the segmentation masks. Thanks to the ease of region-level discrimination in the first stage, the foreground with clear boundaries can be effectively identified and proceeded in the second stage of fine-grained prediction, eliminating false positives within the background. Regarding the anomaly scoring, we adopt histogram statistic of segmentation for logical anomalies and multi-level feature comparison for structural anomalies, respectively. We use scale-invariant Mahalanobis distance to fuse the two types of scoring functions. To comprehensively evaluate our method, we also establish a few-shot LAD benchmark based on MvTec LOCO (Bergmann et al., 2022), the first large-scale dataset featuring LA. A variety of learning-based and training-free LAD methods are included in the benchmark. Notably, owing to its training-free characteristic, our method naturally extends to multiple categories via a unified model, in contrast to the previous state-of-the-art PSAD (Kim et al., 2024; Batzner et al., 2024) that requires training a specialized model for each category. The contributions are three-fold:

- We build up a simple yet effective training-free baseline that only leverages CLIP to parse the logical relations within the image through segmentation.

- We introduce a two-stage segmentation pipeline, which utilizes visual coherence and cross-modal alignment with a coarse-to-fine prompting strategy, to harness the CLIP potentials for accurate segmentation.

- Despite its simplicity, our method achieve the state-of-the-art in our established few-shot LAD benchmark.

## 2 RELATED WORK

**Logical Anomaly Detection (LAD).** Industrial anomaly detection primarily focuses on two types of anomalies: structural and logical. In contrast to structural anomalies such as bents and scratches, which locally exhibit conspicuous visual inconsistency against normal patterns, logical anomalies violate the logical constraints *e.g.*, the quantity, spatial layout or composition of objects. The majority of AD methods (Bergmann et al., 2020; Salehi et al., 2021; Huang et al., 2022; Jeong et al., 2023; Li et al., 2024) and widely-used benchmarks (Bergmann et al., 2019; Zou et al., 2022) are biased towards structural anomalies, inapplicable for detecting logical ones, which requires alternative designs for capturing the global dependencies within the image.

With the release of MvTec LOCO dataset (Bergmann et al., 2022), few efforts are devoted to addressing the issue, and can generally be categorized into two streams. The first is the embedding based methods (Bergmann et al., 2022; Batzner et al., 2024) , which train a model to match with the outputs of the other pre-trained one. Specifically, Efficient-AD (Batzner et al., 2024) equips the student-teacher learning framework with an autoencoder to learn the logical constraints of normal images. GCAD (Bergmann et al., 2022) inherits the framework but improves the anomaly scoring by using reconstruction errors and feature distance to address picturable and unpicturable anomalies, respectively. Some other works attempt synthesize pseudo logical anomalies by either utilizing a diffusion model (Dai et al., 2024) or edge manipulation (Zhao, 2024). Alternative methods reason about logical constraints in images through segmentation. ComAD (Liu et al., 2023) performs K-means on the pre-trained DINO features (Caron et al., 2021) to segment images into multiple components, based on which a series of meticulously designed techniques are developed to model metrological features. The follow-up work PSAD (Kim et al., 2024) improves the segmentation precision and granularity by introducing elaborately annotated masks to train a model to segment object parts. While effective, it heavily depends on the well-trained segmentation model subject to massive training images with a few elaborately annotated ones.

More importantly, all the existing methods work within the full-shot setting and require an additional training process, while degenerate significantly in few-shot setting at a risk of over-fitting. Differently, this work especially focuses on the more challenging few-shot setting, in which only a few normal images without any annotations are available at training, and presents a simple yet effective training-free framework that only utilizes the powerful CLIP to detect logical anomalies.

**Few-shot Anomaly detection (FSAD).** FSAD aims at detecting anomalies with only access to a limited number of normal images. RegAD (Huang et al., 2022) sets up a new paradigm that trains a single generalized model for new categories via feature registration. WinCLIP (Jeong et al., 2023) embraces the pre-trained CLIP to identify anomalies through matching the well-designed text prompts with window-based visual features in the shared vision-language space. The recent PromptAD (Li et al., 2024) improves the results by introducing learnable prompts conditioned on the few-shot normal images. Nevertheless, all these methods work on detecting structural anomalies and fail to acquire the logical dependencies within the image, leaving much room for improvement in LAD.

**Open-vocabulary Semantic Segmentation (OVSS).** Different from conventional segmentation methods that are confined to a infinite visual concepts, OVSS endeavors to segment semantic elements of arbitrary categories, with CLIP being the essential impetus for the growth. However, directly applying it to dense prediction tasks is sub-optimal. SCLIP (Wang et al., 2023) attributes the inferiority to spatial misalignment of patch representations caused by vanilla self-attention modules. To address this, SCLIP introduces a novel self-attention mechanism that facilitates covariant visual features. More recent methods have further enhanced spatial consistency by encouraging each patch attend to its neighbours (Hajimiri et al., 2024; Shao et al., 2024), thereby enhancing the localization capabilities. Despite the promising achievements of these advanced OVSS methods, their direct application to LAD still yields sub-optimal results, due to the domain gap between the general-purpose pre-training data and the images in industrial scenarios. To unleash the potentials of CLIP for LAD, we introduce a series of modifications to fully utilize both visual coherence and cross-modal alignment capabilities at inference, ensuring more accurate object/part segmentation.

## 3  METHOD

### 3.1  COARSE-TO-FINE SEGMENTATION

The main concept for LAD is to analyze the relationships among objects/parts through the precise segmentation. Though CLIP demonstrates strong generalized abilities in zero-shot classification and segmentation, directly adapting it to industrial images yields false positive predictions. We attribute the imprecise segmentation to two key factors: 1) individual foreground patch token at the last ViT stage probably carries ambiguous semantics infiltrated by background; 2) vision-language alignment at patch level is imperfect due to the CLIP pre-training on matching global image embeddings with captions. In this paper, we argue that with proper modifications in CLIP inference, it can precisely identify objects of interest. Inspired by chain-of-thought (CoT) in natural language process (Wei et al., 2022), which involves providing a series of intermediate reasoning steps to guide the model in solving complex problems, we devise a coarse-to-fine segmentation pipeline to derive a precise fine-grained segmentation mask. This approach operates in a top-to-bottom fashion, considering both visual spatial size and textual prompt granularity.

**Foreground Extraction.** While several training-free OVSS methods achieve impressive segmentation results on popular benchmarks such as ADE-20K (Zhou et al., 2019), which primarily feature natural images with rich semantic content, their direct application to industrial images often leads to noisy predictions due to the significant domain gap. Specifically, these incorrect predictions frequently arise around object boundaries or background regions (Fig. 3). This issue is exaggerated when matching visual patch tokens with finer text prompts. The observation inspires us to first extract the foreground region and then perform the fine-grained segmentation within the foreground. Benefiting from the strong visual coherence in shallow ViT stages, we propose to aggregate patches based on their visual similarities. Specifically, given $k$ normal images $I \in \mathbb{R}^{k \times H \times W \times 3}$, we first use CLIP ViT to extract their visual representations $\mathbf{F}_i \in \mathbb{R}^{k \times \frac{H}{s} \times \frac{W}{s} \times C}$ at $i$-th ViT stage, where $s$ denotes patch size and $C$ is the embedding dimension. We omit the layer index $i$ for simplicity. We then perform K-means to obtain $N$ cluster centers $\mu_i \in \mathbb{R}^C (i = 1, 2, ..., N_c)$. Each patch is assigned to the nearest $\mu_i$, forming distinct regions that serve as mask proposals $m_i \in \{0, 1\}^{\frac{H}{s} \times \frac{W}{s}}$. Note that we equally treat each patch regardless of its position. These proposals are used to aggregate patch tokens at the last stage of ViT, which has proven capable of vision-language alignment in addition to the [CLS] token (Zhou et al., 2022). We average the masked-out patch tokens to acquire the representation of the corresponding region $m_i$.

$$\overline{\mathbf{F}}_{m_i} = \mathrm{AvgPool}(\mathbf{F}, m_i) = \frac{\sum \mathbf{F} \odot m_i}{\sum m_i}, \tag{1}$$

where $\odot$ denotes the element-wise multiplication and $\sum$ sums over all the positions of elements. Foreground can be easily identified by matching the region-level representation with a pre-set target embeddings. Here, we provide a set of coarse prompts, which can generally describe the semantics of the regions, *e.g.*, `background or connector`. These words are extended by an ensemble of prompt template, *e.g.*, `a photo of [c]`, which consistently boost the performance (Radford et al., 2021). Compared to common OVSS methods directly performing fine-grained segmentation, binary classification at region level is easy to complete, ensuring more accurate foreground extraction.

**Fine-grained Alignment.** Thanks to the strong visual coherence, we obtain tight boundaries around the foreground with minimal false positives. However, the initial foreground mask is class-agnostic and does not fully capture the compositional relationships among objects. To address this limitation, we incorporate fine-grained segmentation at patch level. Similar with the previous stage, we prepare a set of detailed prompts such as `splicing connector` and `red/yellow/blue cable` for per-patch matching. We disregard any predictions on the background, focusing fine-grained segmentation exclusively within the foreground region. Despite the simplicity of this coarse-to-fine segmentation strategy, the resulting segmentation mask features clear boundaries with precise fine-grained object part masks.

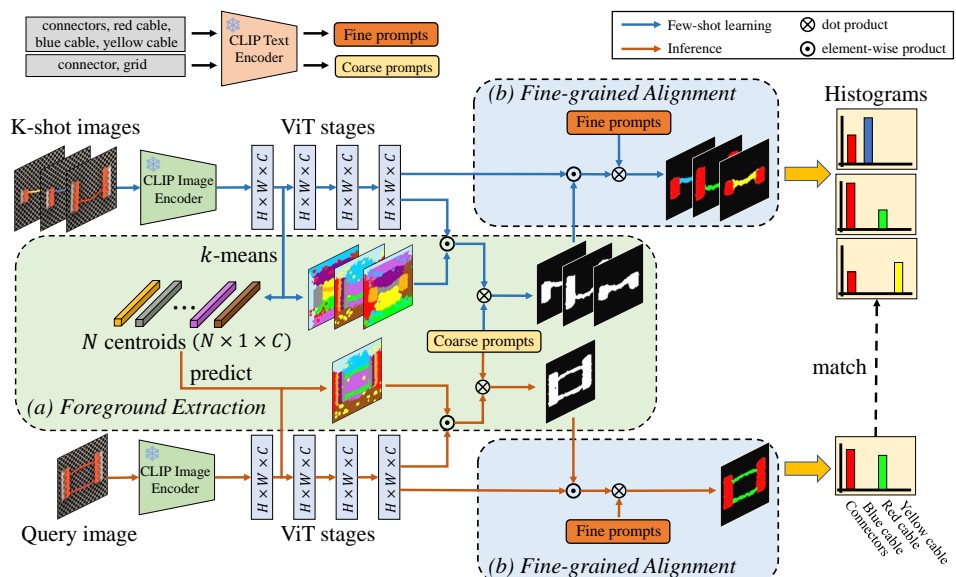

Figure 2: The proposed cross-modal coarse-to-fine segmentation for LAD involves two stages: 1) coarse masks are generated through unsupervised clustering, where averaged patch embeddings are used as mask representation and matched with coarse prompts to extract the foreground; 2) patch embeddings are then matched with fine-grained prompts to classify each patch. The final segmentation masks are created by fusing the coarse and fine masks, from which the class histograms are derived for LAD.

## 3.2 ANOMALY DETECTION

**Inference.** On the LAD, we calculate the class histogram $h_i$ based on the segmentation mask for each of the $k$ normal images and maintain a memory bank $\mathcal{M}_{hist} = \{h_i\}_{i=1}^k$. The number of bins in the histograms corresponds to the number of classes of interest, as indicated by the text prompts, excluding the background class. This histogram memory is assumed to reveal the compositional relationships among objects within the images. Moreover, we reuse the cluster centroids $\mu_i$ derived from the $k$-shot normal images to partition a query image $I_{test}$, which helps make logical anomalies more recognizable through low-level feature matching. We then apply the same coarse-to-fine vision-language alignment used for processing the normal images to obtain the histogram $h_{test}$. While high-level compositional relations are crucial for LAD, investigating low-level appearance differences is essential for effective structural anomaly detection. To the end, we adopt per-patch comparison in feature space to spot the visual defects. Specifically, we first extract multi-level ViT features $\mathbf{F}_i$ for all $k$ images and save patch features $f_j \in \mathbb{R}^C$ at all positions to create a patch memory $\mathcal{M}_{patch} = \{f_j\}_{j=1}^{k \times \frac{H}{s} \times \frac{W}{s}}$. The anomaly map is obtained by comparing the query patch $f_{patch}$ with the those in $\mathcal{M}_{patch}$. Multi-level feature comparison is enabled for robust detection following (Salehi et al., 2021; Wang et al., 2021).

**Anomaly Scoring.** Based on the constructed memories $\mathcal{M}_{hist}$ and $\mathcal{M}_{patch}$, we inspect the difference between a query image and its nearest samples in the $\mathcal{M}_{hist}$ and $\mathcal{M}_{patch}$ to measure the anomaly scores. On the logical anomalies, we adopt the histogram difference ratio as the anomaly scoring, different from directly calculating their the $L_2$ distance (Kim et al., 2024) in consideration of the unbalanced class distribution.

$$s_{log} = \min_{h \in \mathcal{M}_{hist}} \|h_{test}/h\|_1. \tag{2}$$

On the structural anomalies, we select the maximum value from the anomaly map as the resulting anomaly score.

Table 1: Image-level detection comparison on few-shot MvTec LOCO (Bergmann et al., 2022) and the micro-averaged AUROC scores are measured. We report the mean and standard deviation over five random seeds. The best results are boldface.

| Setup | Category | Learning-based | | | | Training-free | | | | | |
|---|---|---|---|---|---|---|---|---|---|---|---|
| | | PromptAD | RD4AD | EfficientAD | STFPM | CLIP | PatchCore | ComAD | WinCLIP | SINBAD | CLIP-LAD |
| 1-shot | Breakfast Box | 72.8±2.1 | 70.7±6.9 | 60.6±1.4 | 59.4±1.6 | 65.0±0.2 | 73.4±1.8 | 69.3±3.2 | 47.6±3.0 | 70.0±2.5 | **74.6**±2.2 |
| | Juice Bottle (Banana) | 77.3±1.5 | 65.8±4.2 | 85.0±3.0 | 88.3±3.5 | 76.6±2.3 | 76.4±0.4 | 72.3±2.8 | 57.6±2.0 | 80.2±4.0 | **88.5**±0.9 |
| | Juice Bottle (Cherry) | 75.7±0.4 | 71.8±2.4 | **88.2**±3.7 | 80.6±2.4 | 64.8±1.3 | 77.3±4.2 | 59.0±5.2 | 72.1±5.5 | 86.5±3.5 | 72.8±1.5 |
| | Juice Bottle (Orange) | 71.4±2.8 | 67.5±0.6 | **81.8**±5.6 | 77.8±4.9 | 60.6±4.2 | 72.9±4.6 | 58.2±1.2 | 60.4±1.6 | 72.7±3.0 | 77.7±3.1 |
| | Pushpins | 72.3±1.9 | 61.0±2.1 | 61.1±1.8 | 58.8±3.4 | 64.8±0.6 | 67.7±2.6 | 64.8±2.8 | 53.1±5.4 | 52.9±3.2 | **82.4**±1.7 |
| | Screw Bag | 53.4±3.3 | 46.6±3.1 | 44.2±2.3 | 51.2±3.2 | 58.2±5.0 | 63.7±3.4 | 54.2±2.7 | 55.8±2.6 | 56.3±2.2 | **72.2**±3.4 |
| | Connectors (Red cable) | 65.3±1.2 | 55.0±4.6 | 68.1±5.6 | 54.2±0.6 | **71.6**±3.3 | 66.2±2.6 | 78.7±0.6 | 50.9±0.0 | 72.1±2.7 | 60.2±1.4 |
| | Connectors (Blue cable) | 72.4±3.8 | 75.7±4.9 | 76.4±4.0 | 59.3±5.7 | 65.0±2.1 | 73.9±3.4 | 60.5±2.2 | 51.8±3.2 | 71.6±2.1 | **78.6**±2.9 |
| | Connectors (Yellow cable) | 67.5±0.9 | 62.9±0.9 | 69.9±1.5 | 60.9±2.1 | 67.5±2.0 | 64.6±1.1 | 73.6±5.9 | 60.4±2.4 | **72.5**±1.6 | 59.3±4.4 |
| | **Average** | 69.8±0.8 | 64.1±1.2 | 70.6±2.2 | 65.6±1.6 | 66.0±1.2 | 70.7±0.7 | 65.6±1.5 | 56.6±1.5 | 70.5±0.6 | **74.0**±1.0 |
| 2-shot | Breakfast Box | 74.3±1.6 | 69.8±3.6 | 62.3±4.8 | 63.8±2.4 | 68.3±2.0 | 72.4±2.1 | 64.7±2.7 | 51.2±2.4 | 75.9±3.0 | **81.1**±2.7 |
| | Juice Bottle (Banana) | 77.7±2.6 | 83.1±6.3 | 87.8±1.1 | 84.7±1.7 | 80.6±2.3 | 76.2±1.3 | 76.6±1.9 | 70.0±4.2 | 84.3±2.5 | **89.7**±3.2 |
| | Juice Bottle (Cherry) | 78.4±1.2 | 76.9±3.0 | **90.4**±2.9 | 89.4±1.6 | 69.9±2.5 | 82.4±2.5 | 68.8±3.8 | 70.5±2.0 | 88.3±4.3 | 83.6±1.2 |
| | Juice Bottle (Orange) | 72.0±2.9 | 74.0±0.4 | **88.3**±2.6 | 84.5±2.4 | 83.8±3.8 | 78.8±1.8 | 64.6±3.9 | 62.0±4.1 | 76.8±1.4 | 77.5±3.9 |
| | Pushpins | 70.4±1.4 | 64.3±3.9 | 58.7±1.3 | 58.7±3.2 | 62.7±4.2 | 71.7±2.1 | 56.8±3.8 | 54.5±2.7 | 51.5±1.2 | **82.1**±2.3 |
| | Screw Bag | 56.0±1.8 | 57.6±0.6 | 44.5±3.9 | 54.2±1.7 | 50.1±3.5 | 64.4±2.2 | 59.4±1.6 | 59.3±0.5 | 55.5±2.1 | **75.4**±7.1 |
| | Connectors (Red cable) | 68.9±3.0 | 54.8±3.3 | **75.8**±1.4 | 66.2±1.2 | 72.6±3.5 | 72.0±3.3 | 73.1±4.2 | 47.6±4.9 | 73.2±5.9 | 69.8±1.7 |
| | Connectors (Blue cable) | 75.9±1.7 | 65.1±4.8 | 79.3±3.4 | 64.3±2.0 | 68.3±1.4 | 76.5±0.2 | **85.1**±3.1 | 60.1±1.3 | 78.0±3.2 | 82.9±4.0 |
| | Connectors (Yellow cable) | 71.7±2.3 | 65.6±1.5 | 70.2±1.7 | 62.0±0.8 | 73.9±2.3 | 72.6±1.3 | **77.9**±4.3 | 61.0±3.1 | 69.5±4.2 | 67.7±3.0 |
| | **Average** | 71.7±1.0 | 67.9±0.7 | 73.0±0.9 | 69.8±0.8 | 70.0±2.2 | 74.1±1.1 | 69.7±1.0 | 59.6±1.3 | 72.6±1.2 | **78.9**±0.8 |
| 4-shot | Breakfast Box | 77.3±1.1 | 66.3±2.7 | 61.5±1.6 | 67.3±5.4 | 74.1±1.7 | 76.4±0.6 | 65.7±2.0 | 57.4±1.9 | 78.0±5.7 | **83.9**±1.7 |
| | Juice Bottle (Banana) | 79.1±2.9 | 87.7±1.9 | 92.3±1.9 | 89.7±1.9 | 82.4±3.3 | 75.6±3.7 | 79.2±3.3 | 66.8±3.3 | 85.9±2.1 | **95.3**±2.1 |
| | Juice Bottle (Cherry) | 76.9±0.9 | 93.4±1.8 | 95.0±2.4 | **95.5**±6.2 | 74.8±3.9 | 84.8±3.9 | 70.0±4.7 | 70.3±2.1 | 85.9±1.8 | 87.1±2.6 |
| | Juice Bottle (Orange) | 75.5±3.2 | 84.2±1.0 | **90.1**±5.7 | 84.9±6.2 | 72.1±0.8 | 82.1±2.1 | 81.9±3.5 | 71.8±1.6 | 77.1±2.8 | 78.2±1.3 |
| | Pushpins | 73.6±1.5 | 65.2±2.7 | 65.0±1.5 | 64.6±1.2 | 65.8±2.1 | 70.4±4.2 | 72.5±3.1 | 51.2±0.8 | 55.3±3.0 | **79.2**±1.1 |
| | Screw Bag | 59.3±4.2 | 56.9±4.2 | 50.4±1.0 | 57.6±3.1 | 56.6±5.7 | 66.1±2.2 | 64.7±3.8 | 54.3±2.2 | 56.1±2.9 | **75.6**±3.7 |
| | Connectors (Red cable) | 63.2±2.9 | 64.2±1.0 | 76.8±1.8 | 73.0±1.8 | 69.4±4.1 | 70.1±4.5 | 66.0±2.8 | 58.0±1.8 | **77.4**±0.6 | 70.3±1.7 |
| | Connectors (Blue cable) | 79.4±2.7 | 76.8±2.0 | 82.5±3.9 | 62.8±4.6 | 73.8±2.8 | 74.3±1.5 | 82.8±4.3 | 64.8±2.8 | 78.9±2.2 | **84.2**±4.3 |
| | Connectors (Yellow cable) | 73.6±1.3 | 63.6±4.4 | 69.7±4.1 | 69.0±1.9 | 65.9±2.4 | 71.7±4.8 | **79.8**±2.0 | 66.5±3.2 | 74.8±0.9 | 77.0±0.3 |
| | **Average** | 73.1±1.6 | 73.1±0.3 | 75.9±0.6 | 73.8±1.5 | 70.5±1.3 | 74.6±1.5 | 73.6±1.9 | 62.3±1.1 | 74.4±2.1 | **81.2**±0.5 |

$$s_{str} = \max_{f \in \mathcal{M}_{patch}} \|f_{test} - f\|_2. \tag{3}$$

**Score Normalization.** We adopt segmentation analysis and patch comparison to cater for detecting logical and structural anomalies, respectively. However, the two strategies generate anomaly scores $s_{log}$ and $s_{str}$ with different scales. Thus, it is essential to reasonably normalize the two types of scoring before aggregation. Following (Batzner et al., 2024), we use the validation set, which contains anomaly-free images and has no overlaps with the training set, to estimate the scales of the two scoring types. With the assumption that each type of anomaly scoring follows a Gaussian distribution, we derive the mean $\mu$ and covariance matrix $\Sigma$ for the set of anomaly scores $(s_{log}, s_{str})$ across the validation images. For an input tuple $x$, we define the Mahalanobis distance as the resulting anomaly score:

$$s(x; \mu, \Sigma) = \sqrt{(x - \mu)\Sigma(x - \mu)^T}. \tag{4}$$

## 4 EXPERIMENT

**Benchmark Set-up.** To the best of our knowledge, MvTec LOCO (Bergmann et al., 2022) is the only large-scale dataset featuring logical anomalies. It contains 3,644 images across five categories from industrial inspection scenarios. The training and validation sets consist solely of anomaly-free images, while the test set contains both anomaly-free images and anomalous images. Typical logical anomalies include missing or surplus objects, and misplacement. To comprehensively evaluate our method based on the dataset, we first establish a few-show FAD benchmark, considering 1/2/4-shot normal images only for training. However, since the categories *juice bottle* and *splicing connectors* in the dataset have multiple sub-types, randomly sampling $k$ images constituting the training set could cause label ambiguity. For instance, *juice bottle* includes three types of liquid-orange, cherry and banana juice-that are all considered normal. The sampled $k$-shot training images may not necessarily include all these sub-types. To eliminate the ambiguity in determining normality or abnormality, we split these two categories into three sub-types each, resulting in a total of nine categories. The area under the ROC curve (AUROC) is used as evaluation metrics. For each $k$-shot setting, we randomly select $k$ normal images across five random seeds. The models are evaluated on both SA and LA detection together, *i.e.,* micro-average. We also provide separate results in

Fig. 2. Following (Kim et al., 2024), we consider image-level AUROC, as logical anomalies are context-based and typically pertain to the entire image.

**Comparison Methods.** Our few-shot LAD benchmark considers a variety of methods: training-free and learning-based. Training-free methods include CLIP (Radford et al., 2021) PatchCore (Roth et al., 2022), WinCLIP (Jeong et al., 2023), ComAD (Liu et al., 2023), SINBAD (Cohen et al., 2023). The learning-based ones are PromptAD (Li et al., 2024), RD4AD (Deng & Li, 2022), STFPM (Wang et al., 2021) and EfficientAD (Batzner et al., 2024). Notably, since all these comparison methods are developed for full-data scenarios, we adapt them to our few-shot setting by using their official implementations to train the models with only the given $k$ normal images.

**Implementation Details.** We use OpenCLIP's implementation[1] with DataComp-1B (Gadre et al., 2024) and CLIP ViT-L/14 for all experiments. Images are resized to $448 \times 448$. Following (Chen et al., 2023), we evenly divide the visual encoder into four stages and apply K-means clustering on embeddings from the first two stages. We adopt NACLIP (Hajimiri et al., 2024), a top-tier training-free method for OVSS, and up-scale feature maps by $\times 2$. Histogram statistics are calculated on the $64 \times 64$ segmentation masks. We find that the widely-used mask refinement techniques, such as DenseCRF (Krähenbühl & Koltun, 2011) and PAMR (Araslanov & Roth, 2020) yield similar results. We use the same prompt templates as (Radford et al., 2021) and ensemble text embeddings. For the complex *juice bottle*, we perform region-level fine-grained classification in Fig. 5 to avoid noisy patch-level results.

**Experimental Results.** Tab. 1 shows the overall comparison with both learning-based and training-free methods in 1/2/4-shot settings. While learning-based methods achieve descend results in some categories, such as *juice bottle* which is consistently positioned in the center of the image, they all fail in more challenging categories like *pushpins* and *screw bag*, where objects are placed randomly. This failure is due to the risk of over-fitting on the limited number of normal images.Training-free methods attempt to leverage the generalized capabilities of pre-trained foundation models; however, they typically perform feature matching at either patch-level (Roth et al., 2022) or image-level (Cohen et al., 2023), which fails to capture object relationships adequately, leading to inferior detection. In contrast, our method, with tailored designs that fully harness the potential of CLIP, significantly outperforms these alternatives.

In Fig. 3, we visualize the segmentation results of NACLIP (Hajimiri et al., 2024), as well as the coarse and the fine stages of our method. Obviously, NACLIP with fine-prompts tends to make incorrect prediction scattered across the image,particularly around object boundaries and in background regions, whereas our method provides more precise segmentation masks. This improvement can be attributed to our coarse-to-fine cross-modal alignment design. The coarse stage involves unsupervised clustering on shallow embeddings, which results in clearer class boundaries but lacks discriminative labels. The second stage focuses on per-pixel prediction using fine text prompts. By combining the results from these two stages, we achieve precise segmentation results that are well-suited for inferring composition relationships for LAD. Additionally, with more training images available, we observe an increase in average performance. To further understand the performance gain, we separately calculate the metrics for logical and structural anomalies, which in Tab. 2 confirms our method's superiority for LAD.

Table 2: Logical and structural detection are evaluated separately on the few-shot MvTec LOCO. The AUROC scores are averaged across all categories over 1/2/4-shot settings.

|  | Methods | Structural | Logical | Average |
|---|---|---|---|---|
| Learning-based | PromptAD | 65.0 | 71.9 | 68.5 |
|  | RD44D | 64.6 | 71.5 | 68.1 |
|  | EfficientAD | 75.9 | 71.8 | 73.9 |
|  | STFPM | 69.2 | 70.3 | 69.8 |
| Training-free | CLIP | 68.7 | 70.1 | 69.4 |
|  | PatchCore | 68.5 | 76.2 | 72.4 |
|  | ComAD | 60.6 | 75.4 | 68.0 |
|  | WinCLIP | 60.6 | 58.8 | 59.7 |
|  | SINBAD | 68.9 | 77.9 | 73.4 |
|  | CLIP-LAD | **77.0** | **81.4** | **79.2** |

---

[1]https://github.com/mlfoundations/open_clip

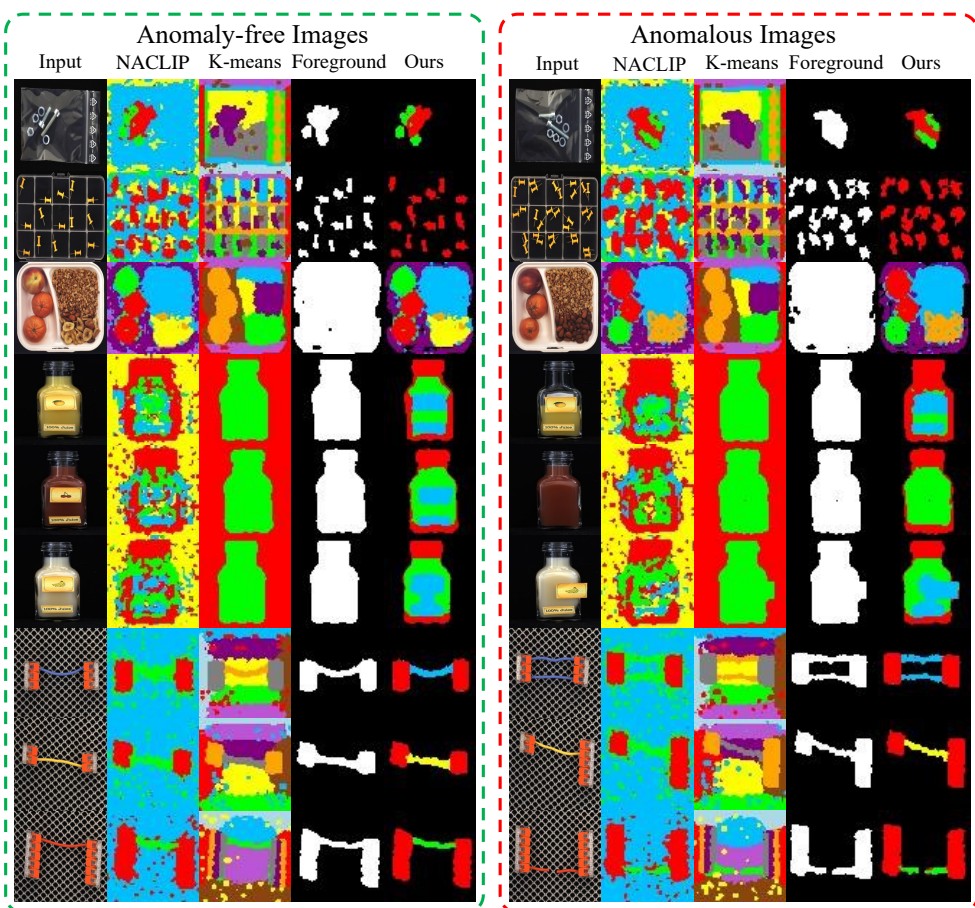

Figure 3: Visualizations of different segmentation methods are shown, displaying the segmentation results of both anomaly-free and anomalous images, each sized $64 \times 64$. The state-of-the-art OVSS method, NACLIP (Hajimiri et al., 2024) suffers from noticeable noisy prediction at arbitrary positions, while our method offers more precise segmentation. The first stage extracts the foreground with clear boundaries by matching the clustered regions with coarse prompts. The second stage performs fine alignment, targeting pixel-level discrimination using fine text prompts, which results in fine-grained segmentation.

Table 3: Logical anomaly detection in multi-class scenario on categories *juice bottle* and *splicing connectors* with only one image per sub-type for training. The AUROC scores are averaged over 1/2/4-shot settings.

|  | Methods | Juice Bottle | Connectors | Average |
|---|---|---|---|---|
| Learning-based | PromptAD | 73.4 | 71.2 | 72.3 |
|  | RD44D | 72.3 | 63.0 | 67.7 |
|  | EfficientAD | 74.8 | 69.2 | 72.0 |
|  | STFPM | 74.1 | 64.5 | 69.3 |
| Training-free | CLIP | 69.1 | 70.9 | 70.0 |
|  | PatchCore | 71.0 | 73.3 | 72.2 |
|  | ComAD | 59.8 | 74.8 | 67.3 |
|  | WinCLIP | 66.5 | 62.2 | 64.4 |
|  | SINBAD | 73.7 | 74.5 | 74.1 |
|  | CLIP-LAD | **81.2** | **78.2** | **79.7** |

**Extension from One-class to Multi-class Scenario.** Recall that in our main experiments, we split the *juice bottle* and *splicing connectors* categories from MvTec LOCO into three sub-types, ensuring that the provided $k$-shot images contain only one class. However, our methods is also applicable to

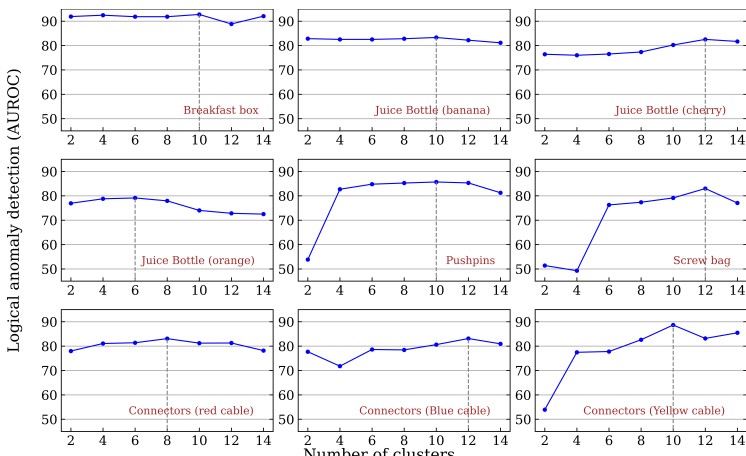

Figure 4: Ablation studies on the number of clusters across nine categories from MvTec LOCO in 4-shot setting. The optimal number for each category is marked.

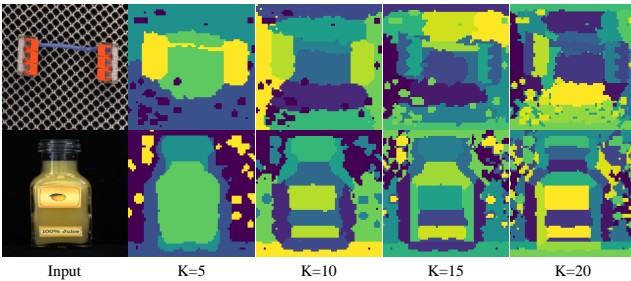

Figure 5: Inappropriate cluster numbers result in either over-clustering or under-clustering.

the multi-class scenario, where the training set consists of multiple classes. In this case, we follow the original normality/abnormality division, treating images that violate the logical constraints of any sub-type as anomalous. We consider the challenging 3-shot setting, with only one image per sub-type. As shown in Tab. 3, our method stills outperforms counterparts significantly, confirming its flexibility in addressing both one-class and multi-class scenarios.

**Effect of Number of Clustering Centers.** The success of our methods lies in its segmentation precision. To achieve precise zero-shot image segmentation, we propose a coarse-to-fine cross-modal alignment. In the coarse stage of foreground extraction, we perform unsupervised clustering on the visual features to aggregate similar patches. The segmentation quality depends largely on the number of clustering centers used. Fig. 4 displays the effects of different cluster numbers across five categories. Over-clustering leads to a large number of fragmented regions, each inevitably mixed with background noise, making it difficult to accurately identify the foreground regions of interest. On the other hand, under-clustering directly results in unclear object boundaries. Thus, it is essential to select an appropriate cluster number for better aggregate the foreground regions.

## 5 CONCLUSION

In this work, we consider the challenging few-shot logical anomaly detection and present a simple yet effective training-free method CLIP-LAD. We leverage the off-the-shelf vision-language model to comprehensively understand the logical constraints through segmentation. Specifically, we devise a cross-modal coarse-to-fine segmenting strategy to take full advantage of visual coherence and cross-modal alignment capabilities in CLIP, facilitating precise segmentation. Experimental results on the few-shot LAD benchmark demonstrate the superiority of our method.

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
