# OpenReview forum: "CLIP-LAD: Unleash the Potential of CLIP for Few-shot Logical Anomaly Detection"
_ICLR.cc/2025/Conference — ICLR 2025 Conference Withdrawn Submission_

### Official Review · Reviewer_WteE · 2024-10-21

**Soundness:** 2
**Presentation:** 2
**Contribution:** 2
**Rating:** 3
**Confidence:** 4

**Summary:**

This paper proposed to utilize CLIP for few-shot anomaly detection. Images are first divided into regions by clustering CLIP's low level features. The regions are first segmented into foreground/background by comparing the region-level representation with a set of coarse prompts. The foreground region is further segmented into semantic classes. Class histogram is computed based on the semantic segmentation results. Anomaly score is computed based on the histogram difference as well as per-patch comparison using multi-level features.

**Strengths:**

* The method shows advantage over other methods on MVTEC LOCO dataset under few-shot settings.

**Weaknesses:**

* Novelty and technical contribution are limited,  as the proposed method is mostly a simple combination of existing techniques, e.g., K-Means clustering, hierarchical clustering that segment foreground first and then segment foreground objects later,  open vocabulary segmentation by aligning visual and text features, anomaly scoring by comparing test sample with normal sample in memory banks.
* The proposed method requires providing detailed prompts for fine-grained segmentation, which incurs additional annotation efforts. The proposed method is essentially performing weakly-supervised segmentation with fine-grained semantic labels provided for each image, while other methods, e.g., EfficientAD, PatchCore, do not require such labels, making the comparison unfair.

**Questions:**

1. Is the number of clusters for each category fixed or varied according to Figure 4?
2. How to generate the detailed prompts needed for fine-grained alignment? How does the quality of such prompts affect the performance?
3. In Eq.(2), the histogram difference ratio is not minimized when the two histograms are identical. E.g., (0.3,0.3,0.4) obtain a score of 3 for identical histogram, and a smaller score of 2.5 for a completely different distribution (0,0,1). As the histogram difference ratio is used as anomaly score, it should be minimized when the test sample has a perfect match in the memory bank. Please clarify.
4. In Ln.268, the paper claimed that "we select the maximum value from the anomaly map as the resulting anomaly score", but in the following Eq.(3), the max is taking over the memory bank. Please clarify.

---

### Official Review · Reviewer_5pgk · 2024-11-01

**Soundness:** 2
**Presentation:** 3
**Contribution:** 2
**Rating:** 5
**Confidence:** 3

**Summary:**

This paper proposes a simple and train-free logical anomaly detection method that operates in a few-shot manner. CLIP-LAD leverages the fine-grained text semantics of CLIP to segment object parts, and store the derived class histograms as memory. An image patch with a significant histogram discrepancy is detected as a logical anomaly.

**Strengths:**

1、Compared to structural anomaly detection, logical anomaly detection remains relatively under-explored. This paper proposes a simple and easy-to-implement solution.

2、The experimental results seem to demonstrate the effectiveness of the proposed method.

3、The paper is generally well-written, with clear organization and effective visual aids.

**Weaknesses:**

1、This paper lacks adequate discussion of the existing anomaly detection literature. As the authors illustrated, structural anomaly detection has attracted much research interest. However, important advancements in anomaly detection, such as few-shot approaches like WinCLIP and InCTRL, as well as zero-shot methods including WinCLIP and AnomalyCLIP, are not mentioned in Introduction section. Furthermore, given that CLIP-LAD is a CLIP-based method, it should engage with previous research that uses CLIP for anomaly detection, e.g., WinCLIP, AnomalyCLIP, and so on. A separate section for CLIP-based anomaly detection in Related Work is suggested for a thorough overview of previous research.

2、From my perspective, the main contribution of this work lies in the design of a fine-grained memory compared to previous memory-based approaches. This fine-grained memory stores class-level histograms derived from clustering path features. While the experimental results demonstrate effectiveness, I am curious about the challenges the authors faced during this process and how they addressed them.
In the current version, the work lacks significant insights or technological innovation that resonate with me. The authors should emphasize their technological contributions more clearly to meet the high publication standards of ICLR.

3、All evaluations were conducted on MvTec LOCO, which has a clear background. This may be a contributing factor to the performance of CLIP-LAD. Additional experiments on more challenging datasets, such as DigitAnatomy or through noise injection into the background of MVTec LOCO, should be conducted to further demonstrate the robustness and superiority of CLIP-LAD.

**Questions:**

See Weaknesses

---

### Official Review · Reviewer_NXqN · 2024-11-06

**Soundness:** 2
**Presentation:** 3
**Contribution:** 1
**Rating:** 1
**Confidence:** 4

**Summary:**

This paper presents a CLIP based method to segment logical objects in images under the setting of few shot learning. The main component is a coarse-to-fine segmentation process, for foreground extraction and fine-grained alignment.
Experiments show some good results.

**Strengths:**

A simple pipeline that works reasonably well. Each component of the overall pipeline is simple and easy to understand

**Weaknesses:**

1) Novelty is very limited. While I do believe novelty is not compulsory for a paper to get accepted to ICLR, I do not see anything I do not know after reading the paper. What is the key point of the method? What information does the paper want to deliver to the readers?
I do not see contribution of the paper
2) While the pipeline is simple, what part of the pipeline contributes most  to final performance?
This is unclear
3) To hardness pretrained CLIP has already generated tons of papers. What  is the main contribution of this paper? What is the difference of the proposed compared to prior work?  Or this paper just introduces CLIP to the task of anomaly detection? This is not clear.
4) The authors claimed a new task termed "logical anomaly detection" is introduced. As the name suggests, I'd expect some reasoning process will be used to perform high-level reasoning for detecting anomlies, I do not see that in the current paper. Actually it's confusing to see the difference between the so-called logical anomaly detection and the standard anomaly detection in the literature

**Questions:**

see the weakness part

---

### Note · Authors · 2024-11-13

I have read and agree with the venue's withdrawal policy on behalf of myself and my co-authors.